# Laboratory Based Surveillance of HIV-1 Acquired Drug Resistance in Cameroon: Implications for Use of Tenofovir-Lamivudine-Dolutegravir (TLD) as Second- or Third-Line Regimens

**DOI:** 10.3390/v15081683

**Published:** 2023-08-02

**Authors:** Joseph Fokam, Collins Ambe Chenwi, Desire Takou, Maria Mercedes Santoro, Valere Tala, George Teto, Grace Beloumou, Ezechiel Ngoufack Jagni Semengue, Beatrice Dambaya, Sandrine Djupsa, Etienne Kembou, Nounouce Pamen Bouba, Rogers Ajeh, Giulia Cappelli, Dora Mbanya, Vittorio Colizzi, Francesca Ceccherini-Silberstein, Carlo-Federico Perno, Alexis Ndjolo

**Affiliations:** 1Chantal BIYA International Reference Centre for Research on HIV/AIDS Prevention and Management (CIRCB), Messa, Yaoundé P.O. Box 3077, Cameroon; dtakou@yahoo.com (D.T.);; 2Faculty of Medicine and Biomedical Sciences (FMBS), University of Yaoundé I, Yaoundé P.O. Box 1364, Cameroon; 3National HIV Drug Resistance Working Group (HIVDRWG), Ministry of Public Health, Yaoundé P.O. Box 3038, Cameroon; 4Faculty of Health Sciences, University of Buea, Buea P.O. Box 063, Cameroon; 5Department of Experimental Medicine, Faculty of Medicine and Surgery, University of Rome “Tor Vergata”, Via Montpellier 1, 00133 Rome, Italy; 6World Health Organisation, Country Office, Yaoundé P.O. Box 155, Cameroon; kemboue@who.int; 7Department of Disease, Epidemic and Pandemic Control, Ministry of Public Health, Yaoundé P.O. Box 3038, Cameroon; 8Central Technical Group, National AIDS Control Committee, Yaoundé P.O. Box 2005, Cameroon; 9Italian National Research Council, P. le Aldo Moro, 7, 00185 Rome, Italy; 10National Blood Transfusion Service, Ministry of Public Health, Yaoundé P.O. Box 3038, Cameroon; 11Haematology and Transfusion Service, Centre Hospitalier et Universitaire (CHU), Yaounde-13, Yaoundé P.O Box 30335, Cameroon; 12Bambino Gesu’ Children’s Research Hospital, Piazza S. Onofrio 4, 00165 Rome, Italy

**Keywords:** laboratory-based surveillance, acquired drug resistance, tenofovir-lamivudine-dolutegravir, TLD, first-line failure, second-line failure

## Abstract

Increased HIV drug resistance (HIVDR) with antiretroviral therapy (ART) rollout may jeopardize therapeutic options, especially in this era of transition to fixed-dose tenofovir-lamivudine-dolutegravir (TLD). We studied acquired HIVDR (ADR) patterns and describe potentially active drugs after first- and second-line failure in resource-limited settings (RLS) like Cameroon. A laboratory-based study with 759 patients (≥15 years) experiencing virological failure was carried out at the Chantal Biya International Reference Centre (CIRCB), Yaoundé, Cameroon. Socio-demographic, therapeutic and immunovirological data from patient records were analysed according to HIV-1 genotypic profiles. Median (IQR) ART-duration was 63 (50–308) months. Median CD4 and viremia were 153 (IQR:50–308) cells/mm^3^ and 138,666 (IQR:28,979–533,066) copies/mL, respectively. Overall ADR was high (93.4% first-line; 92.9%-second-line). TDF, potentially active in 35.7% of participants after first-line and 45.1% after second-line, suggested sub-optimal TLD-efficacy in second-line (64.3%) and third-line (54.9%). All PI/r preserved high efficacy after first-line failure while only DRV/r preserved high-level efficacy (87.9%) after second-line failure. In this resource-limited setting (RLS), ADR is high in ART-failing patients. PI/r strategies remain potent backbones for second-line ART, while only DRV/r remains very potent despite second-line failure. Though TLD use would be preferable, blind use for second- and third-line regimens may be sub-optimal (functional monotherapy with dolutegravir) with high risk of further failure, thus suggesting strategies for selective ART switch to TLD in failing patients in RLS.

## 1. Introduction

Human immunodeficiency virus/acquired immune deficiency syndrome (HIV/AIDS) remains a public health priority, with more than 84.2 million people infected and about 40.1 million related deaths since discovery [1]. Sub-Saharan Africa (SSA) bears the highest burden with two thirds of infected cases, an estimated 25.6 million people [2]. Given improved disease management over the years, current disease prevalence in Cameroon is 2.7% [3].

Following the global 90-90-90 targets set up in 2014 by the World Health Organisation (WHO)/UNAIDS, successful scaleup of antiretroviral therapy (ART) coverage worldwide brought 27.1 million people on treatment by the end of 2017, with a national coverage of 53.68% in Cameroon [4,5]. This increased scalability in ART has been accompanied by numerous benefits, reflected by a progressive and significant decrease in AIDS-related deaths [6,7]. This has however been accompanied by an increase in HIV drug resistance (HIVDR), constituting a major obstacle to HIV management (particularly in the absence of a cure and of an effective vaccine, therefore aiming at life-long treatment), with the potential to jeopardize future treatment options [8]. Acquired HIVDR (ADR) greatly varies worldwide, ranging from 50% in Eswatini to 97% in Uganda for resistance to NNRTI (non-nucleoside reverse transcriptase inhibitor)-based first-line therapy. In Cameroon, the prevalence of drug resistance is 79.86% in patients failing first-line ART, with 38.64% harbouring resistance to all three classes after second-line regimen [9,10]. Given the major risks of HIVDR, the World Health Organisation currently recommends laboratory-based surveillance of HIVDR on remnant viral load samples [11].

Elsewhere, there have been optimizations of the therapeutic guidelines over the years, aimed at further improving treatment outcomes in people living with HIV (PLWHIV). As such, the 2019 WHO guidelines recommend the use of the fixed-dose combination tenofovir-lamivudine-dolutegravir (TLD) as the preferred standard of care in adolescents and adults in first-and third- line regimens [12]. With the high rates of ADR in our context, nucleoside/nucleotide reverse transcriptase inhibitors (NRTI) following transition to TLD may be sub-optimal. Recent findings describe good treatment outcomes and durability of the single-tablet regimen (STR) dolutegravir/lamivudine. Our present study targets patients with virological failure who might be suitable in future for the design of such studies focusing on STR. In addition, because this STR is not yet standard practice in our context, monitoring TLD-treated patients in the presence of high rates of NRTI resistance in the frame of triple ART remains fundamental. Nonetheless, patients with good treatment outcomes might serve in designing future cohorts to evaluate the efficiency of dual-therapy in our context [13,14,15]. Though recent studies show good short-term virological response (mostly measured as viremia < 1000 copies/mL) on TLD despite NRTI (nucleoside/nucleotide reverse transcriptase inhibitor) resistance, the long-term virological response, representing the only major clinical achievement, is not clearly documented as of now [16,17]. Identifying individual candidates for this regimen in our context thus remains essential to better monitor long-term treatment outcomes.

We sought to ascertain patterns of acquired HIVDR in routine clinical practice in Cameroon between 2011 and 2019 and to describe potentially active drugs post-therapeutic failure and their adequacy with the potential use of TLD as a second- or third-line option in this resource-limited setting.

## 2. Materials and Methods

### 2.1. Study Design and Target Population

We carried out a laboratory-based cross-sectional and analytical study from November 2011 to August 2019 with adults (≥15 years) who had failed first- and second-line ART, received for genotypic resistance testing (GRT) at the Chantal Biya International Reference Centre (CIRCB) for Research on HIV and AIDS Prevention and Management, Yaoundé, Cameroon.

### 2.2. Study Site Description

Our study was carried out at CIRCB, Yaoundé, Cameroon and included patients presenting for resistance testing from all ten regions of the country. Located in the capital city, Yaoundé, and created in 2006 with much expertise in the field of HIV/AIDS research, CIRCB offers GRT for PLWHIV at a subsidized cost.

### 2.3. Sample Size Calculation and Sampling Method

Using an estimated prevalence for HIVDR of 94%, as described by Kouamou et al. [18], and a 5% error margin at 95% confidence interval (CI) and taking into account our eight-year period, we obtained a minimal sample size of 696 participants. Participants were recruited following consecutive and exhaustive sampling.

### 2.4. Selection Criteria

Inclusion criteria: We included all participants aged ≥ 15 years, failing virologically (two consecutive plasma viral loads ≥ 1000 copies/mL after intensified adherence counselling) on a standard first- or second-line ART regimen after being treated for at least six months. Of note, adherence was measured using a self-reported method evaluated by the responsible clinician during clinical appointment. All participants had performed genotypic resistance testing at CIRCB.

Exclusion criteria: We did not include patients who were naïve to ART or re-initiating ART (population suitable for pre-treatment HIVDR surveillance), those with undocumented ART therapies, on TB treatment (due to risk of drug interaction) or those with incomplete sequences.

### 2.5. Enrolment Procedure

Participants were selected following our eligibility criteria from a pool of patients in the CIRCB antiviral resistance (CARe) database. Their socio-demographic, clinical and immunovirological data, sequences and GRT results were obtained.

### 2.6. Procedure for GRT

Enrolled patients had previously benefited from GRT following an in-house protocol as follows:

First, 10 mL of blood was drawn from treatment-failing patients who presented for GRT at CIRCB. Plasma samples were then separated from whole blood and stored between −20 °C to −80 °C when samples were not to be analysed immediately. Viral RNA was extracted using a commercially available protocol after an initial two-hour centrifugation of 1 mL of plasma at 14,000 rpm to improve viral RNA concentration and thus increase PCR sensitivity. Following this step, 140 µL of plasma was retained for HIV-1 RNA extraction following the QIAGEN protocol (QIAamp^®^ DNA Minikit; QIAGEN, Courtaboeuf, France). Using an in-house protocol and two-step polymerase chain reaction-PCR (RT-PCR and Nested-PCR), HIV-1 pol-gene (protease and reverse-transcriptase regions) was then amplified (~1600 base pairs). Using agarose gel electrophoresis, the effectiveness and integrity of PCR products was confirmed, following which the amplicons were purified. Purified products were submitted to a sequencing reaction. Products of the sequencing reaction were then gel-purified using SEPHADEX (Seperation Pharmacia Dextran; Cytiva Sweden AB, SE-751 84 Uppsala, Sweden) gel to remove excess reactants and render DNA products pure. Sequencing was performed using an Applied Biosystems 3500 genetic analyzer (HITACHI, Tokyo, Japan) and sequence assembly and editing with SeqScape v2.7 (Applied Biosystems) or Recall v2.28 [19]. The Stanford University database (Stanford HIVdb) was used to identify drug resistance mutations (DRMs).

### 2.7. Sequence Interpretation and Analysis

All edited sequences were interpreted using Stanford HIVdb v.9.4. to identify major DRMs and potentially active medications. Rapid subtyping tools, such as COMET (https://comet.lih.lu/, accessed on 21 June 2020) and REGA (http://dbpartners.stanford.edu:8080/RegaSubtyping/stanford-hiv/typingtool/, accessed on 21 June 2020), were used for subtyping, with confirmations performed using molecular phylogeny after sequence alignment with BioEdit v.7.2.5 and tree inference with MEGA software v.70.107. Sequences generated for the purpose of this study were submitted to Genbank under the following accession numbers: MK580543-MK580618; JF273935-JF273965; JQ796152-JQ796169, MK867721-MK867755; and OQ985493-OQ985958.

### 2.8. Data Interpretation and Statistical Analysis

Data was entered and analysed using Epi Info version 7.2.2.6. Quantitative variables were described using mean (standard deviation) and median (interquartile range). Qualitative variables were described with frequencies and proportions stating 95% confidence intervals (CI). Associations between qualitative variables were calculated using chi square and Fischer’s exact tests where appropriate. Student’s T test was used to compare means of continuous variables between groups, and *p*-values < 0.05 were considered statistically significant. Any *p* < 0.2 from bivariate analysis was included for multivariate analysis through logistic regression, adjusting for potential confounders such age, sex, region or HIV-1 subtypes.

### 2.9. Ethical Considerations

Ethical clearance was sought and obtained from the Institutional Review Board of the Faculty of Medicine and Biomedical Sciences, University of Yaoundé I (reference number 330/UYI/FMSB/VDRC/DAASR/CSD). Research authorisation was also obtained from the CIRCB directorate to carry out our study in this establishment. Confidentiality was ensured by using patient identification numbers with password-protected computers, and core ethical values were respected.

## 3. Results

### 3.1. Socio-Demographic and Clinical Data

We retained a total of 759 participants with mean age (±SD) of 42 (±14) years. Majority of participants were female: 485/745 (63.9%). All 10 regions of the country were represented, with participants from the centre: *n* = 436 (57.4%); northwest: *n* = 86 (11.3%); and littoral: *n* = 85/745 (11.2%) regions. Of the 759 participants, 575 (75.8%) were failing non-nucleoside reverse transcriptase inhibitor (NNRTI)-based first-line ART, while 184 (24.2%) were failing protease inhibitor (PI/r)-based second-line (NB: The second-line failing patients had previously failed first-line regimens as recommended by national treatment guidelines). Table 1 shows a more detailed therapeutic history of the participants.

### 3.2. Immuno-Virological Characteristics

Overall, median (IQR) CD4 cell count at the time of failure was low; 153 (50–308) cells/µL. With respect to the therapeutic lines, immunological status was similar between first- and second-line treated patients (median CD4: 161 (56–315) cells/µL for first-line and 126 (39–299) cells/µL for second-line, *p* = 0.18).

As to virological status, the overall median viral load (IQR) (global) was high at 138,666 (28,979–533,066) copies/mL. Similarly to immunological status, there was no significant difference in the median viral load (IQR) between first- and second-line patients at 134,645 (29,461–555,654) copies/mL and 142,710 (28,331–507,564) copies/mL, respectively (*p* = 0.81; Table 1).

### 3.3. Acquired Drug Resistance Patterns

Globally, overall drug resistance rate (presence of at least one major DRM) was high at 708/759 (93.3% CI; 91.3–94.9%). Prevalence of drug resistance by antiretroviral class were as follows: nucleoside/nucleotide reverse transcriptase inhibitor (NRTI) resistance: 651/759 (85.8%); NNRTI resistance: 685/759 (90.3%); and protease inhibitor (PI/r) resistance: 105/759 (13.8%). Dual-class resistance was as follows: NRTI_NNRTI: 83.3% (632/759); NRTI_PI/r resistance: 13.4% (102/657); and NNRTI_PI/r resistance: 12.4% (94/759). Triple class resistance was 12.4% (94/459). Overall, the most prevalent mutation was M184V (75.5%), followed by K103N (47.4%) and Y181C (27.7%). See Figure 1a,b.

After first-line failure, overall HIVDR (presence of at least one major DRM) was 93.4% (CI 91.1–95.2%) and varied by drug class as follows: NRTI resistance: 87.7%; NNRTI resistance: 92.2%; PI/r resistance: 5%. NRTI_NNRTI dual class resistance: 86.7%. After second-line failure, overall-HIVDR was 92.9%, (95% CI 88.1–96.1%) with 80.0% resistance to NRTI, 41.8% resistance to PI/r, and 40.1% NRTI_PI/r dual-class resistance. The most prevalent PI/r resistance mutation was M46I (29.8%), followed by I54V (14.9%) and I84V (12.8%). See Figure 1c.

The bar charts to the left of Figure 1a–c show the prevalences of major mutations by drug classes: the yellow, blue and red bars describe mutation prevalence irrespective of treatment regimen (overall), after first-line failure, and after second-line failure, respectively. The dot plots to the right of the figure show the frequencies of each mutation. The yellow, blue and red dots represent frequencies irrespective of treatment regimen (overall), after first-line failure, and after second-line failure, respectively.

### 3.4. Genetic Diversity

We observed a total of 36 clades. The major viral clade was CRF02_AG, making up 60.3%, (56.8–63.8%) of the viral subtypes, followed by A_1_ (6.1%; 5.6–8.0%) then G (5.7%; 4.2–7.5%). Pure subtypes made up 24.2% of the viral subclades, while recombinant subtypes made up 75.8%. We did not find any association between HIV-1 viral clade and CD4, viral load or HIVDR occurrence (Table 2). Thus, rate of resistance, viral load and CD4 count are not dependent on the clade/subtype of HIV-1; rather they are linked to the clinical status and the therapeutic regimens used.

### 3.5. Potentially Active Drugs

#### 3.5.1. Potential Efficacy of Commonly Used Antiretroviral Drugs after First- and Second-Line Failure

We considered effective drugs as those having a susceptibility score of <30 according to the Stanford HIVDR database. After first-line failure, irrespective of NRTI exposure, zidovudine (AZT) was potentially active in 46.1% (95% CI: 42.1–50.2%) of participants; tenofovir (TDF) in 35.7% (95% CI: 31.9–39.7%); abacavir (ABC) in 18.5% (95% CI: 15.6–21.9%); and lamivudine/emtricitabine (3TC/FTC) in 8.7% (95% CI: 10.5–21.5%). As to NNRTI, efavirenz (EFV) was potentially active in 13.2% of participants (95% CI; 10.7–16.2%) and nevirapine (NVP) in 7.6%, (95% CI; 5.7–10.1%) of participants (Figure 2).

After second-line failure, AZT appeared potentially active in 35.2% (95% CI: 28.3–42.6%) of participants; tenofovir (TDF) in 45.1% (95% CI: 37.7–52.6%); ABC in 22.5% (95% CI: 16.7–29.3%); and 3TC/FTC in 15.4% (95% CI: 10.5–21.5%). Darunavir (DRV/r) appeared potentially active in 86.7% (84.8–94.8%) of participants, unlike atazanavir (ATV/r) and lopinavir (LPV/r) which appeared potentially active in 61.0% (95% CI 63.5–68.1%). See Figure 2.

#### 3.5.2. Efficacy of NRTI Molecules with Respect to NRTI Exposure

After AZT/D4T + XTC exposure in first-line treatment, few participants had virus susceptible to AZT (30.7%), TDF (41.0%) and ABC (21.0%). On the contrary, after TDF + XTC exposure, 63.1% of participants had viral susceptibility to AZT, 30.9% to TDF and 18.4% to ABC (See Figure 3). Note that mutation K65R, typically present at TDF (and often ABC) failure, increases susceptibility to AZT, thus making the latter a relevant option in patients failing TDF/ABC.

#### 3.5.3. Efficacy of ATV/r, LPV/r and DRV/r with Respect to PI/r Exposure

DRV/r preserved good levels of efficacy despite PI/r history. After ATV/r and LPV/r exposure, DRV/r preserved efficacy in 96.6% and 81.0% of participants, respectively. After both ATV/r and LPV/r exposure, it preserved efficacy in 87.8% (75.2–95.4%) of participants, particularly if used at 600 mg bid. Lastly, after previous exposure to other PIs (NFV and IDV), it preserved efficacy in 60.0% of individuals (Figure 4).

#### 3.5.4. Predictive Efficacy of TLD

Taking into consideration the potential efficacy of TDF after first- and second-line failure (35.7% and 45.1% respectively), and considering the non-exposure to integrase strand transfer inhibitors (and thus the potential efficacy of dolutegravir), full TLD efficacy was predicted in 35.7% and 45.1% of patients after first- and second-line failures, respectively. With respect to drug exposure, predictive efficacy of TLD appeared lowest in patients with prior exposure to TDF-only regimens, followed by those exposed to both AZT/D4T and TDF, and highest in those exposed to AZT-only regimens (30.9%, 36.3% and 41.0%, respectively). Lastly, after second-line failure, TLD predictive efficacy appears highest in those who had received only AZT/D4T-containing regimens (50.0%), followed by those who had received TDF-only regimens (47.0%) and those who had received both TDF and AZT/D4T (35.1%).

## 4. Discussion

The scalability of ART comes with expected increases in HIVDR in a resource-limited setting (RLS). Our study aimed at describing the patterns of HIVDR in patients failing ART in Cameroon between 2011 and 2019. Overall rates of ADR were high in these patients, with 93.3% presenting with at least one major DRM. These results are not surprising as HIVDR constitutes the major reason for patients failing ART in Cameroon. The development of these resistances in the clinical context could be favoured by poor adherence to ART, stock ruptures, sub-therapeutic blood levels of drug (probably due to poor absorption, drug-drug interactions or food-drug interactions), poor prescription practices and intrinsic viral factors, amongst others [20,21]. In addition, the long duration of treatment (median duration ~5 years) indicates heavily treated patients with the increased possibility of accumulating DRMs over time. Elsewhere, the poor immunovirological status in this population at the time of genotypic resistance testing (median, CD4 count 153 cells/mm^3^ and median viral load, 138,666 copies/mL) suggests late detection of failure, which could also explain the high levels of ADR observed. Moreover, the high viral load (5log) and low CD4 count observed after first- and second-line failure in our context is different from values obtained in western countries (moderate level viremia that is in the order of 3log) but more like in patients with advanced failure and higher risk of clinical progression and death [22,23]. Taking U=U (undetectable=untransmissible) into consideration, the high viral loads and high rates of resistance at failure in these patients increase the risk of HIV transmission, with an increased risk of transmitting drug resistances [24,25,26]. There is therefore a need for better and more effective treatment in these patients, given that the risks of transmission, clinical progression and death are higher. This further advocates treatment guided by resistance testing because we cannot afford another failure [27]. In this RLS, where resistance tests may not be available for all patients, the question of genotyping for treatment initiation should not be a major concern. As a matter of fact, in an era where NNRTI-based therapies (weak genetic barriers) are being phased out and replaced by DTG-based regimens, baseline drug resistance to DTG is very low (overall integrase resistance: 0.8%; integrase resistance in ART-naïve: 0.0%; integrase resistance in ART-experienced: 0.9%) [28]. With this low rate of DTG resistance across an ART-naïve population, GRT might not be essential for treatment initiation. This is particularly true because resistance to TDF and 3TC are also very low at ART initiation. In addition, DTG-based regimens overcome the natural resistance of some HIV clades (such as HIV-1 group O, group N and HIV-2). Nevertheless, we recommend continuous surveillance of resistance at population level and the use of resistance testing in heavily treated, treatment-failing patients due to the possible accumulation of drug resistance mutations [29]. HIV-1 genetic variability was high in this population, with a total of 36 identified viral clades led principally by recombinant subtypes (75.8%), notably CRF02_AG (60.3%). Although this high genetic variability reiterates the rich biodiversity of HIV in Cameroon, harbouring all HIV-subtypes, it also underscores the difficulties that may accompany the management of this disease in our context and probably contribute to the high levels of resistance observed.

In patients failing first-line therapy, rates of ADR were high (93.4%), with similar trends across the drug classes with 97.7% for NRTI, 92.2% for NNRTI and 86.7% for NRTI_NNRTI dual-class resistance. These results are similar to those obtained in Taiwan by Tsai et al. in 2017 and Manasa et al. in South Africa [30,31]. This supports the argument that HIVDR is the main factor responsible for treatment failure in our context. As for PI/r resistance, rates after first-line failure were, not surprisingly, low (5%), supported by the absence of PI/r drug pressure in first-line treated patients. The PI/r resistance in first-line patients would thus be driven mostly by the presence of transmitted drug resistances, which is low in the PI/r class, as described in Mexico by Gerardo et al. in 2018 who found the prevalence of PI/r transmitted drug resistance to be 4.9% [32]. After second-line failure, rates of NRTI and NNRTI resistance all remained high (92.9% and 84.1% respectively). PI/r resistance was higher (41.8%) than in first-line failure, as in previous studies. Nevertheless, this result suggests the possibility of recycling PI/r for further therapeutic options even after second-line failure in about half of failing patients [33], also considering the relatively limited cross-resistance of darunavir (particularly if used at 600 mg bid) after the development of resistance to other PIs.

As concerns potentially active regimes after first-line failure, AZT preserved efficacy in 46% of patients, TDF in 35.7% and ABC in 18.5%. In this era of transition to TLD, this highlights the risk of sub-optimal therapy in about 65% of patients, given the poor efficacy of TDF (DTG functional monotherapy). Though many studies have shown good efficacy of TLD despite NRTI resistance, clinical monitoring of this group of patients is crucial in order to evaluate the long-term efficacy of such regimens and also permit opportune detection of failure [16,17]. This also advocates the better and greater use of viral load for the early detection of virological failure in first- and second-line regimens, given that the accumulation of mutations occurs heavily in patients where the virus replicates in the presence of non-effective drugs and drug concentrations; early detection of failure spares the accumulation of mutations and increases the chance of efficacy of the next regime. In addition, in this context of high viral load and low CD4 count at failure, the risk of failure with functional DTG monotherapy is even higher. As concerns NNRTIs, EFV and NVP preserved activity in a very limited proportion of patients (13.2% and 7.6%, respectively). These results are similar to other studies carried out in Africa and can be explained by the fact that these first-generation NRTIs have low genetic barriers with greater propensity for accumulating drug resistance mutations [34,35]. This indicates a very limited need for these drugs in future therapeutic options after treatment failure [36]. Not surprisingly, all commonly used protease inhibitors in our context (ATV/r, LPV/r and DRV/r) preserved high levels of efficacy after first-line failure (94.6%, 94.8% and 96.6%, respectively) [36]. These results highlight the fact that in resource-limited settings like ours, PIs may constitute a driving arm for second-line therapy in first-line failing patients. With respect to NRTI exposure, AZT preserved efficacy in most patients (63.1%) after first line failure with documented exposure to TDF-only regimens. On the contrary, after exposure to AZT/D4T, both TDF and ABC preserved efficacy in a limited number of participants. This is explained by the fact that patients exposed to thymidine analogues (AZT/D4T) may develop thymidine analogue mutations (TAMs), which have variable degrees of cross-resistance to other molecules of the same class (ABC and TDF). As such, ABC and TDF may be sub-optimal in most patients having received AZT/D4T-containing regimens, while AZT remains effective in patients failing TDF- or ABC-only regimens due to the absence of cross-resistance with AZT, unless transmitted. Therefore, in patients with a clearly documented treatment history of TDF-only regimens and no previous treatment with AZT/D4T, blind switches to second-line therapy could be made with a PI/r associated with 3TC and AZT, even in the absence of genotypic resistance testing, with great chance of success [27].

After second-line failure, all commonly used NRTIs in our context (TDF, AZT and ABC) appeared to be effective only in a low proportion of participants (45.1%, 37.7% and 22.5%, respectively). This indicates that care needs to be taken in choosing subsequent NRTI-accompanying drugs following second-line failure with the need for genotypic resistance testing. This also highlights the fact that blind transitions to TLD in patients failing second-line therapy in our context may be sub-optimal in about half of patients. Interestingly, DRV/r remained effective in the majority of patients (96.6%), even after second-line failure and despite the risks of cross-resistance by mutations to ATV/r and LPV/r. Therefore, DRV/r remains recyclable in third-line regimens in our context [37].

## 5. Conclusions

In this resource-limited setting, using WHO guidelines for ART management and laboratory-based HIVDR surveillance, rates of ADR are very high after first- and second-line failures, favoured by long treatment duration and late failure detection. After an NNRTI-based first-line, PI/r and a substituted NRTI-accompanying molecule may be sufficient in second-line treatment, especially with a well-documented treatment history, but may need GRT guidance in cases of the first-line use of thymidine analogues. After second-line failure, DRV/r remains very effective, thus representing a great option for third-line regimens. Lastly, TLD may be sub-optimal (functional monotherapy with DTG) when used blindly as a second- or third-line regimen. This calls for cautious monitoring and use by designing a sequential algorithm for switching heavily treated failing patients to TLD, especially those with previous exposure to TDF-containing regimens. These measures will ensure the long-term efficacy of TLD in resource-limited settings.

## Figures and Tables

**Figure 1 viruses-15-01683-f001:**
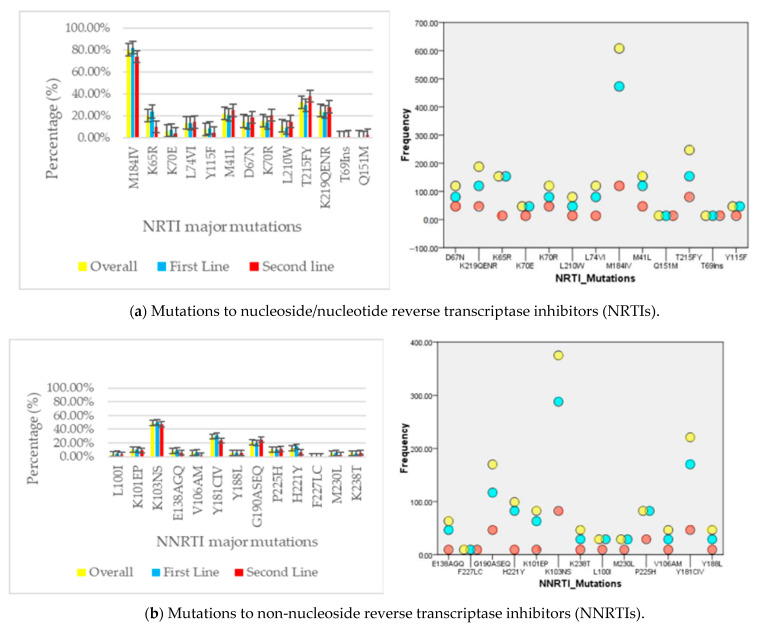
Mutational profiles for (**a**) NRTI, (**b**) NNRTI and (**c**) PI/r resistance.

**Figure 2 viruses-15-01683-f002:**
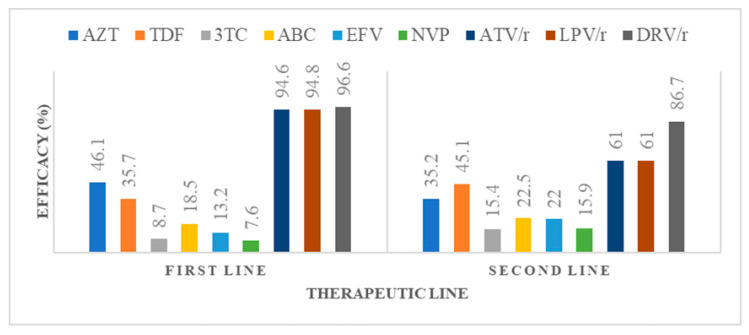
Potential efficacy of commonly used antiretroviral drugs with respect to therapeutic lines. The figure shows the percentage of participants (*y* axis) in whom the listed antiretroviral drugs (*x* axis) appeared effective after first-line (**left**) and second-line (**right**) failure. Note the high levels of effective DRV/r even after second-line failure. AZT: zidovudine; TDF: tenofovir; 3TC: lamivudine; ABC: abacavir; EFV: efavirenz; NVP: nevirapine; ATV/r: atazanavir; LPV/r: lopinavir; DRV/r: darunavir.

**Figure 3 viruses-15-01683-f003:**
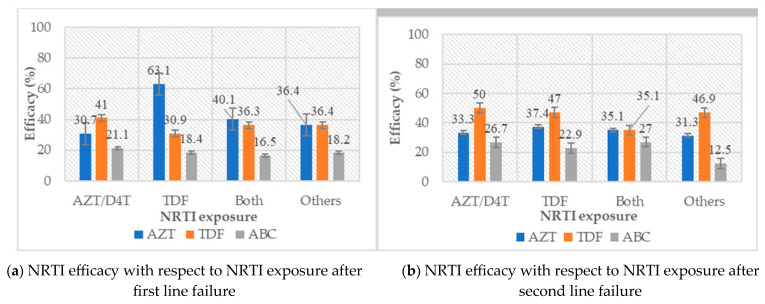
Efficacy of nucleoside/nucleotide reverse transcriptase inhibitors (NRTIs) after (**a**) first- and (**b**) second-line failures. This figure shows the percentage of participants with preserved efficacy (*y* axis) of locally available NRTIs (zidovudine (AZT), abacavir (ABC) and tenofovir (TDF); *x* axis), with respect to NRTI exposure in both first- and second-line failing patients. Note the poor levels of efficacy of ABC and TDF after sole exposure to AZT in both first- and second-line regimens are explained by cross-resistance due to thymidine analogue mutations.

**Figure 4 viruses-15-01683-f004:**
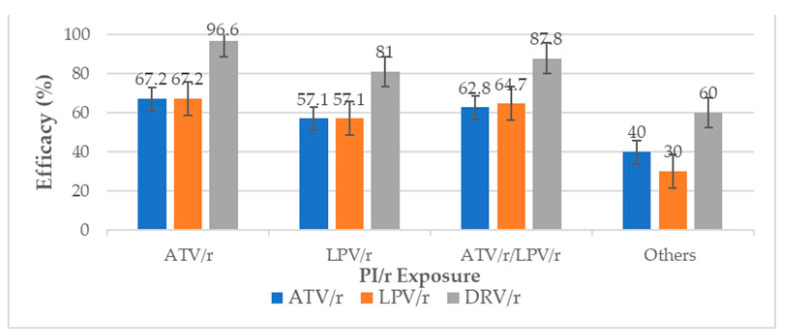
Efficacy of protease inhibitors (PI/r) with respect to PI/r exposure. The figure demonstrates the percentage of participants with preserved efficacy (*y* axis) of PI/r (*x* axis) after PI/r exposure. ‘Others’ refers to older protease inhibitors such as nelfinavir and indinavir. We note that irrespective of the initial PI/r used, viral susceptibility to DRV/r was observed in a good number of second-line failing participants. ATV/r: atazanavir; LPV/r: lopinavir; DRV/r: darunavir.

**Table 1 viruses-15-01683-t001:** Summary of sociodemographic and clinical data of study participants.

Characteristic	All Participants (*n* = 759)
Age in years, mean ± SD	42 ± 14
Gender *n* (%)	
Females	485 (63.9%)
Males	274 (36.1%)
Viral load (copies/mL), median (IQR)	138,666 (28,979–533,066)
CD 4 Count (cells/µL), median (IQR)	153 (50–308)
ART Duration in months, median (IQR)	63 (35.0–105)
Therapeutic exposure, *n* (%)	
First-line ART regimens	575 (75.8%)
First-line NRTI exposure	
- ABC + XTC	1 (0.2%)
- AZT/D4T + XTC	166 (28.8%)
- TDF + XTC	217 (37.6%)
- * AZT/D4T + XTC and subsequently TDF + XTC	182 (31.5%)
- Other (DDI + XTC)	11 (1.91%)
Second-line ART regimens	184 (24.2%)
Second-Line PI/r exposure	
- ATV/r	58 (31.9%)
- LPV/r	63 (34.6%)
- ATV/r and subsequently LPV/r	51 (28.0%)
- Other (IND, NFV)	10 (5.5%)
Second-line NRTI exposure	
- AZT/D4T + XTC	30 (16.5%)
- TDF + XTC	83 (45.6%)
- * AZT/D4T + XTC and subsequently TDF + XTC	37 (20.3%)
- Other (DDI + XTC)	32 (17.6%)

IQR: interquartile range; SD: standard deviation; ART: antiretroviral therapy; ABC: abacavir; XTC: lamivudine or emtricitabine; D4T: stavudine; AZT: zidovudine; TDF: tenofovir; DDI: didanosine; ATV/r: atazanavir; LPV/r: lopinavir; IND: indinavir; NFV: nelfinavir. * This group refers to participants who had received both AZT/D4 + XTC and subsequently experienced a substitution with TDF + XTC in their treatment histories. Patients on AZT/D4T + XTC, TDF + XTC or ABC + XTC were considered first-line when the backbone molecule was an NNRTI or second-line when the backbone molecule was a PI/r. The AZT/D4T + XTC and subsequently TDF + XTC patients experienced an NRTI substitution within the same treatment line, primarily due to adverse drug events necessitating a change in the NRTI component or occasional drug stock outs requiring NRTI-substitution within the same treatment line.

**Table 2 viruses-15-01683-t002:** Effect of HIV-1 viral clade on immunovirological status and HIV Drug resistance occurrence.

	CRF02_AG	Non-CRF02_AG	OR, *p*-Value	Pure	Recombinant	OR, *p*-Value
Median viral load (copies/mL)	150 (38–285)	158 (55–299)	NA, 0.32	159 (56–301)	150 (38–287)	NA, 0.4
Median CD4 (cells/ µL)	145,872 (23,805–505,458)	132,913 (23,998–416,268)	NA, 0.44	115,979 (29,008–573,478)	149,947 (26,664–467,444)	NA, 0.8
HIVDR (%)	92.8	94.9	0.7, 0.286	96.7	92.6	2.3, 0.055

## Data Availability

Not applicable.

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
