# Peer review of "Laboratory Based Surveillance of HIV-1 Acquired Drug Resistance in Cameroon: Implications for Use of Tenofovir-Lamivudine-Dolutegravir (TLD) as Second- or Third-Line Regimens"

_viruses, 2023, doi:10.3390/v15081683_

Round 1

Reviewer 1 Report

“Laboratory Based Surveillance of HIV-1 Acquired Drug Re-2 sistance in Cameroon: Implications for use of tenofo-3 vir-lamivudine-dolutegravir (TLD) as second- or third-line 4 regimens” from Fokam et al, it´s a very interesting, informative and important paper in the context of drug resistance mutations and management of therapy in resource limited settings.

Scientifically the paper it is good and it´s well writen.

Some minor suggestions to improve clarity are provided below:

Material and Methods:

Please provide sequence GenBank accession numbers.

Results:

- The antiretroviral drugs names mut appear in full the first time they are mentioned in the text as well in the figures legends

- Figure 1a - Please add chart title, and legends to x and y axes.

- Section 3.4 do not exist, please correct this and the next sections

- In section 3.5 add a phylogentic tree in the first paragraph

-In section 3.5, a table summarizing the information in the second paragraph would make it easier to understand

- In figure 3 in the legend it´s mentioned 2A and 2B figures, correct to 3A and 3B Also the charts aren´t identified with A and B, add.

- In section 3.6.3, you mentioned figure 3 in the text but it is figure 4.

In several places I detected extra space between words, please review the text carefully

Author Response

Comments and Suggestions for Authors

“Laboratory Based Surveillance of HIV-1 Acquired Drug Re-2 sistance in Cameroon: Implications for use of tenofo-3 vir-lamivudine-dolutegravir (TLD) as second- or third-line 4 regimens” from Fokam et al, it´s a very interesting, informative and important paper in the context of drug resistance mutations and management of therapy in resource limited settings.

Scientifically the paper it is good and it´s well writen.

Some minor suggestions to improve clarity are provided below:

Material and Methods:

Please provide sequence GenBank accession numbers.

Response: We are grateful with this comment from the reviewer. Accession numbers have been provided and the related statement has been added under the methods section.

Results:

- The antiretroviral drugs names mut appear in full the first time they are mentioned in the text as well in the figures legends

Response: These have been updated in the revised manuscript.

- Figure 1a - Please add chart title, and legends to x and y axes.

Response: These have been updated accordingly in the revised manuscript (Lines 217 to 226).

- Section 3.4 do not exist, please correct this and the next sections

Response: This has been updated accordingly.

- In section 3.5 add a phylogenetic tree in the first paragraph

Response:  We thank the reviewer for this remark. The phylogenetic tree has been added as recommended (Lines 238 to 250).

-In section 3.5, a table summarizing the information in the second paragraph would make it easier to understand.

Response: We thank the reviewer for this proposition which we have taken into consideration in the revised manuscript (lines 257 to 258).

- In figure 3 in the legend it´s mentioned 2A and 2B figures, correct to 3A and 3B Also the charts aren´t identified with A and B, add.

Response: These have been updated accordingly in the revised manuscript (line 298).

- In section 3.6.3, you mentioned figure 3 in the text but it is figure 4.

Response: Updated accordingly in the revised manuscript.

In several places I detected extra space between words, please review the text carefully.

Response: We thank the reviewer for this remark. We have gone through thoroughly and hope to have ameliorated a lot this aspect.  

Submission Date

28 March 2023

Date of this review

24 Apr 2023 18:03:37

Reviewer 2 Report

In this study by Fokam et al., the authors evaluated the effectiveness of different regimens for antiretroviral therapy.  HIV drug resistance as show by the failure to suppress viremia was determined. Interestingly, all commonly used protease inhibitors are highly effective after the failure of first-line ART.  This supports the use of protease inhibitors as an effective second-line treatment against HIV rebound.  Importantly, after second-line failure, nucleoside/nucleotide reverse transcriptase inhibitors (NRTI) are also not highly effective.  While this study is potentially interestingly, data presentation is in general sub-standard.  Figures are not clearly labeled, and statistics analyses are not presented.  Additionally, the writing is difficult to read, with many abbreviations not defined clearly.  Specific comments are as follows:

1. Figures are shown as bar graphs.  The authors should present data as dot plots, with each dot  represent the data from an individual patient.  This will clearly show how many samples are in each group.    

2.  No statistics analysis is shown in any of the figures.

3.  Figures labeling are very unclear.  What does “Chart Title” mean in Figure 1a? What does “y-axis” represent in Figure 1a? Please label the figures and clarify in the figure legend. 

4. Figure 1b: the “Percentage” for y-axis seems to indicate viremia after PI/r treatment.  Please clearly state this in the figure legend. 

5. What does Efficacy (%) mean in Figure 2?  Does it mean the percentage of complete suppression of viremia?  Please clarify this in the figure legend.  Additionally, Y-axis in this figure has no scales. 

6. Please define Efficacy (%) mean in Figures 3 and 4?  

7.  All abbreviations, including drug combinations, should be define when they appear in the text.  

Author Response

Comments and Suggestions for Authors

In this study by Fokam et al., the authors evaluated the effectiveness of different regimens for antiretroviral therapy.  HIV drug resistance as show by the failure to suppress viremia was determined. Interestingly, all commonly used protease inhibitors are highly effective after the failure of first-line ART.  This supports the use of protease inhibitors as an effective second-line treatment against HIV rebound.  Importantly, after second-line failure, nucleoside/nucleotide reverse transcriptase inhibitors (NRTI) are also not highly effective.  While this study is potentially interestingly, data presentation is in general sub-standard.  Figures are not clearly labeled, and statistics analyses are not presented.  Additionally, the writing is difficult to read, with many abbreviations not defined clearly.  Specific comments are as follows:

  1. Figures are shown as bar graphs.  The authors should present data as dot plots, with each dot represent the data from an individual patient.  This will clearly show how many samples are in each group.    

Response: We thank the reviewer for this remark. That notwithstanding, similar data in the literature on HIVDR are also presented as such, making it coherent for comparability of trends and burden. We thus plead with the reviewer to reconsider his view, as the dot plot presentation will show the same trend and information, but may also look over congested, given the great number of participants.

  1. No statistics analysis is shown in any of the figures.

Response: We thanks the reviewer for this comment. Actually, all the figures contain statistics related to the data, except for figures 1a and 1b where the bars indicating the 95% confidence interval of data were duly added. Adding frequencies on figures 1a and 1b would make it congested and difficult to read. To address the reviewer’s comment, the frequencies provided on the y axis could serve as scale in identifying the respective frequencies. We believe this would be suitable for readers of the paper.

  1. Figures labelling are very unclear.  What does “Chart Title” mean in Figure 1a? What does “y-axis” represent in Figure 1a? Please label the figures and clarify in the figure legend. 

Response: We thank the reviewer for this remark. This has been updated in the revised manuscript; see results section (lines 217 to 227).

  1. Figure 1b: the “Percentage” for y-axis seems to indicate viremia after PI/r treatment.  Please clearly state this in the figure legend. 

Response: This has been updated accordingly  (line 228).

  1. What does Efficacy (%) mean in Figure 2?  Does it mean the percentage of complete suppression of viremia?  Please clarify this in the figure legend.  Additionally, Y-axis in this figure has no scales. 

Response: We thank the reviewer for these pertinent remarks. “Efficacy (%)” referred to the percentage of participants in whom drug efficacy was preserved. The definitions have been updated in the figure legends. Values have also been added to figure 2 (now fig 3). (Lines 281 to 286).

  1. Please define Efficacy (%) mean in Figures 3 and 4?  

Response: This has been defined in the updated manuscript.

  1. All abbreviations, including drug combinations, should be define when they appear in the text.  

Response: These have been updated accordingly.

Submission Date

28 March 2023

Date of this review

28 Apr 2023 18:58:54

Reviewer 3 Report

General comments

The abbreviation "ART" is used throughout the text without being defined. It would be helpful to define it the first time it is used. Please recheck the manuscript and fix this issue.

The text would benefit from some editing for grammar and clarity. For example, there are some run-on sentences and awkward phrasing that could be revised for clarity.

Introduction

The introduction could benefit from a more concise summary of the main points. The information about HIV/AIDS prevalence and ART coverage could be summarized in a few sentences to provide context for the study.

The sentence “In Cameroon, the prevalence of drug resistance is 79.86% in patients on 74 first-line and 38.64% in patients on second-line regimen” is unclear. Please reformulate it

I would like more information about the STR with dolutegravir, particularly about the efficacy and durability of this regimens (I suggest some papers that you could read: 10.1089/apc.2021.0089, 10.1097/QAI.0000000000002787 ).

Methods

Regarding the inclusion criteria, you wrote “Non-inclusion: We did not include patients presenting with pre-treatment HIVDR mutations”. However, it is unclear if all the patients performed the GRT before starting the treatment. Please comment.

Results

It is unclear why you add the 95%CI of the percentage of female, regions of provenience, failure exc.

In table 1 the NNRTI used is missing. In addition, I do not understand what “Both AZT/D4T+XTC and TDF+XTC” means. In addition, it is not clear if the HIV-RNA and CD4 count are at the moment of failure or at the moment of starting the first regimens.

The use of round and square brackets is unclear.

The paragraph 3.3 is very confusing and hard to follow. I suggest to simplify it, removing the 95%CI

Figure 2 should be revised. You should add the numbers on the efficacy axes.

It is not clear what “3.5.2. Efficacy of NRTI molecules with respect to NRTI exposure” means.

Also, figure 3 should be revised since in  3A the percentages are 0,20,40,80, and in 3B, 0,10,20,30,40,50,60. In addition, in the caption, you wrote 2A and 2B instead of 3A and 3B.

Figure 4 should be revised. Having 120% is impossible. In addition, the 95%CI fo DRV/r in people exposed to ATVr is higher than 100%.

Discussion

You never used the abbreviation RLS, please write it entirely.

I suggest writing which could be the valid regimens for these patients, for example, DRV/r every 12 hours plus DTG 50mg every 12 hours.

In addition, I would like to read a sentence about U=U and how for these patients, the risk of transmitting the virus is high (10.1097/QAD.0000000000002825, 10.1016/S0140-6736(19)30418-0).

Finally, do you think GRT should also be performed in people before starting the treatment? In Europe, the needing is discussed since there is no difference in terms of outcome between those who performed and those who did not (10.1002/jmv.27754)

Author Response

Comments and Suggestions for Authors

General comments

The abbreviation "ART" is used throughout the text without being defined. It would be helpful to define it the first time it is used. Please recheck the manuscript and fix this issue.

Response: We have fixed this in the updated manuscript (Line 63).

The text would benefit from some editing for grammar and clarity. For example, there are some run-on sentences and awkward phrasing that could be revised for clarity.

 Response: We thank the reviewer again for this pertinent remark which has been raised by almost all reviewers. We have thoroughly gone through the manuscript again, and hope the written language is more comprehensive for readers now.

Introduction

The introduction could benefit from a more concise summary of the main points. The information about HIV/AIDS prevalence and ART coverage could be summarized in a few sentences to provide context for the study.

Response: We have summarized this part as indicated, to make it more concise (lines 56 to 65).

The sentence “In Cameroon, the prevalence of drug resistance is 79.86% in patients on 74 first-line and 38.64% in patients on second-line regimen” is unclear. Please reformulate it

Repose: This has been reformulated for better clarity (Lines 72 to 74).

I would like more information about the STR with dolutegravir, particularly about the efficacy and durability of this regimens (I suggest some papers that you could read: 10.1089/apc.2021.0089, 10.1097/QAI.0000000000002787).

Response: We thank the reviewer for this additional data. We have added this information and the references have been used to strengthen our study rationale (see lines 83 to 90). These papers show good and durable treatment outcomes with STR dolutegravir following a suppressed viremia or in ART-naïve individuals. Our present study is rather targeting patients with virological failure who might be suitable in future for the design of such studies focusing STR. To add, since this STR is not yet standard practice in our context, monitoring TLD treated patients in the presence of high rates of NRTI resistance in the frame of triple ART remains fundamental. Nonetheless, patients with good treatment outcomes might serve in designing future cohorts to evaluate the efficiency dual-therapy with our context.

Methods

Regarding the inclusion criteria, you wrote “Non-inclusion: We did not include patients presenting with pre-treatment HIVDR mutations”. However, it is unclear if all the patients performed the GRT before starting the treatment. Please comment.

Response: We thank the reviewer for this observation. We have rephrased this sentence for better comprehension. As a matter of fact, we did not include ART-naïve patients, or patients re-initiating ART, as this population is rather suitable for PDR surveillance. We therefore hope the revised sentence is clearer for potential readers.

Results

It is unclear why you add the 95%CI of the percentage of female, regions of provenience, failure exc.

Response: We have corrected this in the revised manuscript

In table 1 the NNRTI used is missing. In addition, I do not understand what “Both AZT/D4T+XTC and TDF+XTC” means. In addition, it is not clear if the HIV-RNA and CD4 count are at the moment of failure or at the moment of starting the first regimens.

Response: In table one, the NNRTI drugs used according to tour guidelines were first generation NNRTI (ie NVP or EFV). Elsewhere, we laid emphasis on NRTI therapeutic exposure and PI/r exposure, as these were the drugs that could significantly influence subsequent therapeutic options. Therefore, “Both AZT/D4T+XTC and TDF+XTC” refers to patients who were exposed to these two NRTI accompanying regimes. The NRTI resistance profile here is expected to be mixed (resistance mutations to AZT/D4T and TDF), and therefore different clinical implications.   

The use of round and square brackets is unclear.

Response: Square brackets were used in our manuscript for references, and to indicate interquartile ranges. Elsewhere, round brackets were used (ie either for definitions of abbreviations/acronyms, precision purposes, examples etc).

The paragraph 3.3 is very confusing and hard to follow. I suggest simplifying it, removing the 95%CI

Response: Thank you for this remark. We have revised the paragraph for better understanding.

Figure 2 should be revised. You should add the numbers on the efficacy axes.

Response: We have updated this in the revised manuscript.

It is not clear what “3.5.2. Efficacy of NRTI molecules with respect to NRTI exposure” means.

Also, figure 3 should be revised since in  3A the percentages are 0,20,40,80, and in 3B, 0,10,20,30,40,50,60. In addition, in the caption, you wrote 2A and 2B instead of 3A and 3B.

Response: This has been corrected in the updated manuscript

Figure 4 should be revised. Having 120% is impossible. In addition, the 95%CI fo DRV/r in people exposed to ATVr is higher than 100%.

Response: We have corrected this in the revised manuscript.

Discussion

You never used the abbreviation RLS, please write it entirely.

Response: This has been corrected in the revised manuscript (Lines 330 to 331).

I suggest writing which could be the valid regimens for these patients, for example, DRV/r every 12 hours plus DTG 50mg every 12 hours.

Response: From our evidence, DRV/r efficacy was higher after failure on first-line (96.6%) as compared to those failing second-line (86.7%). This indicates that DRV/r for a public health approach would be more suitable on second-line regimens, while its use in a third-line combination should be driven by genotyping for the appropriate dosing according to the mutational profile. Therefore, we propose not to state treatment dosing in our current discussion and data interpretation as these could be very variable from one treatment line to another. We therefore plead with the reviewer to leave treatment combinations open, as they may greatly vary from patient to patient after therapeutic failure, and with respect to the clinical context.

In addition, I would like to read a sentence about U=U and how for these patients, the risk of transmitting the virus is high (10.1097/QAD.0000000000002825, 10.1016/S0140-6736(19)30418-0).

Response: We thank the reviewer for this interesting way of looking at the results. Please find updates as suggested in the discussion (Lines 348 to 350).

Finally, do you think GRT should also be performed in people before starting the treatment? In Europe, the needing is discussed since there is no difference in terms of outcome between those who performed and those who did not (10.1002/jmv.27754)

Response: We thank the reviewer for this pertinent interrogation. As a matter of fact, in an era where NNRTI-based therapies (weak genetic barriers) are being phased out and replaced by DTG-based regimen in our context, baseline drug resistance to DTG in our context is very low (overall INSTI resistance 0.8%, resistance in ART naïve 0.0%, resistance in ART-experienced 0.9%, see our publication doi:10.1093/jac/dkab004). Since DTG resistance is 0% among ART-naive population, GRT might not be essential for treatment initiation. This is particularly true because the resistance to TDF and 3TC are also very low at ART initiation. To add, DTG-based regimens overcome the natural resistance by some HIV clades (such as HIV-1 group O, group N and HIV-2…). Nonetheless, we recommend continuous surveillance of resistance at population-level, and also the use of GRT in heavily treated, treatment failing patients (see our paper on a case of INSTI resistance at DOI:https://doi.org/10.1186/s13756-020-00799-2).

Submission Date

28 March 2023

Date of this review

01 May 2023 21:37:56

Round 2

Reviewer 2 Report

This comment is not addressed:

1. Figures are shown as bar graphs.  The authors should present data as dot plots, with each dot  represent the data from an individual patient.  This will clearly show how many samples are in each group.   

2.  No statistics analysis is shown in any of the figures.

Author Response

This comment is not addressed:

  1. Figures are shown as bar graphs.  The authors should present data as dot plots, with each dot represent the data from an individual patient.  This will clearly show how many samples are in each group. 

Response: We thank the reviewer for this remark. In the current version of the manuscript, we have added dot plots representations for a complete illustration of the data as required by the reviewer.   

  1. No statistics analysis is shown in any of the figures.

Response: We thank for requiring this clarity. We effectively performed statistical analyses, as well as methodology for multivariate analyses (see the updated statement in the methods section). In table 2 (Effect of HIV-1 viral clade on immunovirological status and HIV Drug resistance occurrence), we provided statistcal analyses), we provided bivariate statistical analyses for which all p-values were greater than 0.2: in this context, no test of statistical analyses were eligible for logistic regresssion. The other data were mainly descritpve as aforemetioned in the methods section.We hope the current version is clearer and that these inputs now meet the reviewer’s suggestion.  

Reviewer 3 Report

Thank you for having a reply to all my comments.

I still do not understand what "Both AZT/D4T+XTC and TDF+XTC" means. How is it possible that in the first regimens, they used both AZT/3TC and TDF/3TC? If they used both, it means that they changed the regimens. The only explanation could be that you defined "change regimens" only when there is a modification of the third drug. 

Figure 2. I do not understand the needing to add this figure. I suggest removing it.

Figure 4. There is an overlapping of the letters.

I like your answer about the GRT use in naive patients. I suggest adding it to the discussion.

Author Response

Thank you for having a reply to all my comments.

I still do not understand what "Both AZT/D4T+XTC and TDF+XTC" means. How is it possible that in the first regimens, they used both AZT/3TC and TDF/3TC? If they used both, it means that they changed the regimens. The only explanation could be that you defined "change regimens" only when there is a modification of the third drug. 

Response: We thank the reviewer for highlighting the need for clarity at this point. In effect, the regimens were administered subsequently. We have provided this clarification in table 1 by adding subsequently in the table (see highlights in yellow). Therefore, the statement "Both AZT/D4T+XTC and TDF+XTC" has been changed to "AZT/D4T+XTC and subsequently TDF+XTC".

Figure 2. I do not understand the needing to add this figure. I suggest removing it.

Response: We thank the reviewer for this remark. This figure was asked by reviewer 1. However, we agree with reviewer 3 that deleting the figure does not affect the results of the phylogenetic analysis, which is also presented in the text (see the result section). Therefore, we have removed Fig 2 as recommended.

Figure 4. There is an overlapping of the letters.

Response: We have corrected this in the revised manuscript.

I like your answer about the GRT use in naive patients. I suggest adding it to the discussion.

Response: We thank the reviewer for this remark. We have added this to the discussion as recommended.